# From genes to reproductive health: Immune cell influences on abortion

**Dan Shen**[1☯], **Wendi Xu**[2☯], **Jingyi Zheng**[3☯], **YiZhou Cao**[4], **Xinyi Bo**[4], **FeiXian Fu**[1], **Bing Wen**[1], **Fuqiang Zhou**[1], **Jing Cao**[1]*

**1** Department of Radiology, Yiyang Central Hospital, Yiyang, Hunan, P. R. China, **2** Department of General Surgery, The Second Xiangya Hospital of Central South University, Changsha, Hunan, P. R. China, **3** Department of Medical Imaging, The Fourth Hosiptal of Changsha, Changsha, Hunan, P. R. China, **4** Graduate Collaborative Training Base of Yiyang Central Hospital, Hengyang Medical School, University of South China, Hengyang, Hunan, P. R. China

☯ These authors contributed equally to this work.
* ffx4791@outlook.com

## Abstract

### Background

The relationship between dysregulation of the immune system and reproductive health, particularly in the context of abortion, is an area of critical research. Identifying the immunological factors that contribute to abortion could provide valuable insights into its prevention and management.

### Methods

This study used bidirectional two-sample Mendelian Randomization (MR) approach to evaluate the causal link between 731 immune cell features and the risk of abortion. The study analyzed GWAS data from 257,561 Europeans, including 7,069 cases and 250,492 controls, by utilizing genetic variation as instrumental variables. The immune phenotypes included several cell types, including B cells, T cells, TBNK cells, Treg cells, and monocytes. These were analyzed using the 'TwoSampleMR' package in R software.

### Results

The study identified 34 immune phenotypes that have a significant causal relationship with abortion risk. Notably, Results from the B cell group showed a positive correlation between abortion and certain phenotypes, including Unsw mem %B cell, PB/PC %B cell, IgD+ CD24 + %B cell and Naive-mature B cell %lymphocyte. In the T cell group, certain maturation stages such as Naive CD8br %T cell and CD4 on CD45RA+ CD4+ exhibited negative causal links, whereas CCR7 on naive CD8br showed a positive association. The group of Treg cells showed both positive and negative causal relationships with abortion, highlighting the complexity of immune regulation in reproductive health.

### Conclusions

This study reflects the causal relationship between different subtypes of different immune cells and abortion. The results underscore the importance of the immune system in

**Data Availability Statement:** The data on Abortion in this paper and the data on immune cells in 731 are from public databases and are publicly available for download, as follows: GWAS Catalog (GCST90018786, GCST90001391~2121): www.

ebi.ac.uk/gwas/. The data on miscarriage is sourced from [https://www.ebi.ac.uk/gwas/publications/34594039]. The data on immune cells is sourced from [https://www.ebi.ac.uk/gwas/publications/32929287]. All data are publicly available for download. Additionally, we have supplemented the data sources in the main text as well.

**Funding:** The author(s) received no specific funding for this work.

**Competing interests:** The authors have declared that no competing interests exist.

reproductive health and suggest potential therapeutic interventions targeting these immuno-logical pathways.

## Introduction

Abortion is a complex process that seriously threatens the health of women of childbearing age [1]. Approximately 15.3% of women have an abortion [2]. Common causes of abortion include anatomical abnormalities of the female reproductive tract, chromosomal abnormalities, endocrine abnormalities, and infections [3]. However, the cause of abortion is still unclear in 40–50% of patients [4]. Most abortions are thought to be related to immunological factors [5]. Therefore, further research on the interaction between abortion and immune factors is necessary.

A variety of immune cells play a crucial role in maintaining immune balance during pregnancy. For example, a study showed that T cells can affect pregnancy outcomes by inducing Th-cells with different functions, such as Th1 and Th2, causing an imbalance of Th1 and Th2 [6]. In addition, it has been revealed that decreased levels of Treg cells can lead to adverse pregnancy outcomes, which can be used as a potential predictor of abortion [7]. Recent research suggests that an imbalance in the ratio between NK1 and NK2 may indicate abortion. The higher the value, the higher the risk of abortion [8].

Various immune factors secreted by different types of immune cell subtypes are crucial in maintaining a balanced immune tolerance at the immune-maternal-fetal interface [9,10]. This has important implications for women's reproductive health. Understanding how immune cell phenotypes influence miscarriage risk is crucial for advancing reproductive medicine and enhancing maternal health outcomes.

Prior research has established that immune cells could play a role in abortion [11,12]. Therefore, our study is based on the hypothesis that certain genetically determined immune cell phenotypes may be associated with the likelihood of abortion. In this case, we adopted a new approach, the epidemiological study design, the Mendelian randomization (MR) Analysis [13]. This approach is a robust way to evaluate the causal link between genetic variation and clinical outcomes. Utilizing genetic variation as instrumental variables, MR Analysis is able to isolate complex associations between exposure and outcomes while reducing the confounding and bias inherent in traditional observational studies [14]. Currently, some studies have used Mendelian randomization as a method in the study of diseases of the female reproductive system and have determined the causal relationship between immune cells and diseases such as polycystic ovary syndrome, endometriosis, and female reproductive disorders [15–17]. This provides new tools for understanding the basis of female reproductive diseases and immunology.

Our analysis includes 731 immune cell signatures, categorizing them into seven distinct cell families: B cells, T cells, TBNK cells, Treg cells, cDC cells, bone marrow cells, and monocytes [18]. These immune cells perform immune functions, from cell counting to surface antigen expression, providing a broad lineage for analysis. Simultaneously, a two-sample Mendelian randomization (MR) analysis was conducted utilizing extensive genome-wide association study (GWAS) data from 257,561 individuals in Europe, encompassing 7,069 abortion cases and 250,492 controls [19]. We used rigorous statistical methods to obtain the results and performed a variety of sensitivity analyses to ensure the robustness of the results. We aimed to elucidate the immune cell phenotype associated with the risk of abortion and to clarify the

directionality of the association. This is essential for individualized interventions, which may significantly improve reproductive health.

In conclusion, this study represents a novel approach in reproductive immunology, leveraging the power of genetic epidemiology to elucidate the connections between immune cell traits and the likelihood of abortion. Our research has the potential to significantly impact clinical practices and improve outcomes for women worldwide by providing insights into the genetic basis of immune-mediated processes in reproduction.

## Materials and methods

### Study design

By employing bidirectional two-sample Mendelian randomization (MR) Analysis, we investigated the causal relationship between abortion and 731 immune cell characteristics, which were grouped into 7 categories. In the realm of causal inference, MR leverages genetic variations as proxies for potential risk factors, with instrumental variables (IVs) needing to satisfy three crucial assumptions for valid causal inference: (1) There was a strong correlation between genetic instrumental variables and the risk factors studied. (2) genetic instrumental variables were not correlated with any confounding factors. (3) genetic instrumental variables only affected outcomes through risk factors. The study analyzed GWAS data from 257,561 Europeans, comprising 7,069 cases and 250,492 controls. The GWAS encompassed 24,139,422 single nucleotide polymorphisms (SNPs) [19].

### Immunity-wide GWAS data sources

Each immune cell trait involved in our study is available in the publicly available GWAS database directory (registration numbers from GCST90001391 to GCST90002121) [20]. The catalog includes 731 immunophenotypes, categorizing them into seven distinct cell families: B cells, T cells, TBNK cells, Treg cells, cDC cells, bone marrow cells, and monocytes. The GWAS for initial immune characterization utilized data from 3,757 Europeans with no overlapping cohorts. Using a Sardinian sequence-based reference panel, SNPs for approximately 22 million high-density array genotypes were calculated. Correlations were analyzed after adjusting for covariates such as sex and age [21].

### Selection of instrumental variables (IVs)

Genetic variation is closely associated with exposure factors. The significance level of the instrumental variable (IV) for immune signatures is typically set at $5 \times 10^{-6}$. Researchers often utilize the 'TwoSampleMR' package data to identify independent instrumental variables and set $R^2 < 0.001$, kb = 10,000 to remove linkage disequilibrium (LD). When investigating the association between genetic factors and abortion, we adjusted the significance level and chose a higher threshold of $5 \times 10^{-8}$ for genome-wide association studies (GWAS). Additionally, the LD threshold was set to $R^2 < 0.001$, kb = 10,000 to uphold the accuracy and reliability of the research findings.

### Statistical analysis

We utilized R software version 4.2.1 for all procedures (http://www.Rproject.org). To investigate the causal relationship between 731 immune phenotypes and abortion, we primarily employed the 'TwoSampleMR' package (version 0.5.7) for MR analysis. In our study, the IVW method was used as the primary statistical method for MR analysis, in addition to the weighted median method as a complementary method, which provides robust causal estimates even

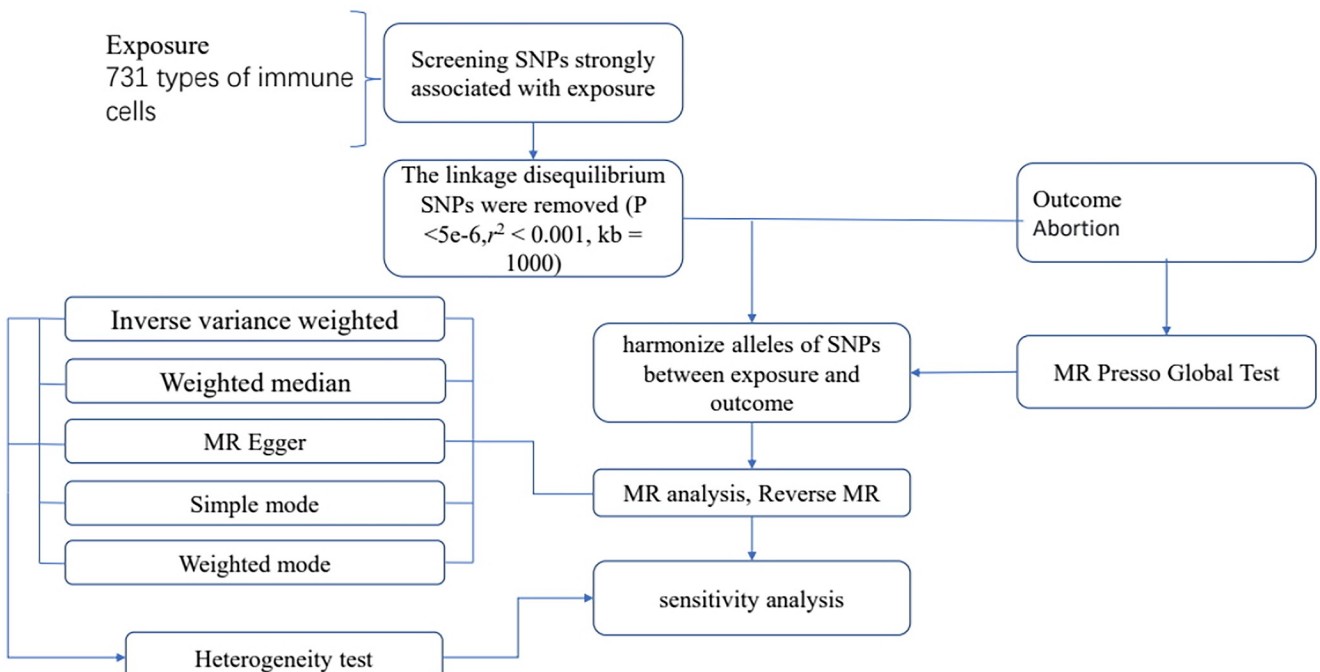

**Fig 1. Flow diagram for quality control of the instrumental variables (IVs) and the entire Mendelian Randomization (MR) analysis process.**
*Abbreviations:** SNPs, single-nucleotide polymorphisms; IVW, inverse variance weighted; MR, Mendelian Randomization; MR Presso, Mendelian Randomization Pleiotropy RESidual Sum and Outlier.

when certain instrumental variables may not be valid if certain assumptions are satisfied. To enhance the robustness and reliability of our findings on the causal relationship between immune phenotype and miscarriage, sensitivity analyses were conducted. Furthermore, Cochran's Q test was utilized to evaluate heterogeneity among instrumental variables. This statistical assessment, based on the comprehensive available data, enhances the accuracy and confidence of our results. The entire process is illustrated in **Fig 1.**

## Results

### Exploration of the causal effect of immunophenotypes on abortion risk

This study identified 34 immune phenotypes that were causally related to the occurrence of abortion ($p < 0.05$), including 5 B cell subtypes, 4 T cell subtypes, 9 TBNK cell subtypes, 7 Treg cell subtypes, 7 cDC cell subtypes, and 2 monocyte subtypes (as shown in **Fig 2** and **S1 Table**). Among the B cell populations identified as being significantly associated with abortion in this study ($p < 0.05$), Unsw mem %B cell, PB/PC %B cell, IgD+ CD24+ %B cell and Naive-mature B cell %lymphocyte were found to increase the risk of abortion (OR > 1), whereas CD20 on IgD- CD27- was associated with a decreased risk of abortion (OR < 1). In Maturation stages of T cell group, Naive CD8br %T cell, CD4 on CD45RA+ CD4+ and CCR7 on naive CD4+ were negatively associated with abortion (OR < 1), while CCR7 on naive CD8br was positively associated with abortion (OR > 1). In TBNK cell group, CD4+ AC was negatively associated with abortion (OR < 1). CD8br %T cell, CD4+ CD8dim AC, CD4+ CD8dim %lymphocyte, CD4+ CD8dim %leukocyte, HLA DR+ CD4+ %lymphocyte, HLA DR+ NK AC, CD45 on lymphocyte and CD45 on CD8br were positively associated with abortion (OR > 1).

| Traits | Methods | pval | OR (95% CI) |
|---|---|---|---|
| Unsw mem %B cell | MR Egger | 0.2862 | 1.0302 (0.9771 - 1.0862) |
| | Weighted median | 0.3973 | 1.0281 (0.9642 - 1.0962) |
| | Inverse variance weighted | 0.0249 | 1.0449 (1.0056 - 1.0858) |
| | Simple mode | 0.1121 | 1.1054 (0.9828 - 1.2433) |
| | Weighted mode | 0.5953 | 1.0161 (0.9589 - 1.0768) |
| PB/PC %B cell | MR Egger | 0.1097 | 1.0454 (0.9927 - 1.1009) |
| | Weighted median | 0.0720 | 1.0509 (0.9956 - 1.1094) |
| | Inverse variance weighted | 0.0121 | 1.0515 (1.0110 - 1.0936) |
| | Simple mode | 0.1934 | 1.0906 (0.9614 - 1.2371) |
| | Weighted mode | 0.0353 | 1.0571 (1.0075 - 1.1090) |
| IgD+ CD24+ %B cell | MR Egger | 0.2677 | 1.0932 (0.9441 - 1.2659) |
| | Weighted median | 0.0144 | 1.1529 (1.0287 - 1.2920) |
| | Inverse variance weighted | <0.001 | 1.1580 (1.0680 - 1.2556) |
| | Simple mode | 0.0682 | 1.1608 (1.0080 - 1.3367) |
| | Weighted mode | 0.0558 | 1.1737 (1.0173 - 1.3540) |
| Naive-mature B cell %lymphocyte | MR Egger | 0.3663 | 1.1792 (0.8369 - 1.6613) |
| | Weighted median | 0.1132 | 1.1226 (0.9729 - 1.2953) |
| | Inverse variance weighted | 0.0235 | 1.1430 (1.0182 - 1.2831) |
| | Simple mode | 0.0522 | 1.3189 (1.0253 - 1.6966) |
| | Weighted mode | 0.0657 | 1.3016 (1.0085 - 1.6799) |
| CD62L- HLA DR++ monocyte %monocyte | MR Egger | 0.0981 | 1.0760 (1.0025 - 1.1549) |
| | Weighted median | 0.0355 | 1.0728 (1.0048 - 1.1454) |
| | Inverse variance weighted | 0.0055 | 1.0672 (1.0193 - 1.1172) |
| | Simple mode | 0.1886 | 1.0809 (0.9753 - 1.1980) |
| | Weighted mode | 0.0517 | 1.0736 (1.0136 - 1.1372) |
| DC AC | MR Egger | 0.2096 | 0.9342 (0.8448 - 1.0331) |
| | Weighted median | 0.0224 | 0.9172 (0.8516 - 0.9878) |
| | Inverse variance weighted | 0.0443 | 0.9424 (0.8895 - 0.9985) |
| | Simple mode | 0.6550 | 0.9722 (0.8614 - 1.0972) |
| | Weighted mode | 0.0615 | 0.9177 (0.8452 - 0.9964) |
| CD62L- DC AC | MR Egger | 0.1299 | 0.9719 (0.9396 - 1.0054) |
| | Weighted median | 0.0243 | 0.9594 (0.9254 - 0.9946) |
| | Inverse variance weighted | 0.0031 | 0.9574 (0.9303 - 0.9854) |
| | Simple mode | 0.0650 | 0.9201 (0.8497 - 0.9964) |
| | Weighted mode | 0.0321 | 0.9622 (0.9329 - 0.9923) |
| CD62L- DC %DC | MR Egger | 0.6212 | 0.9874 (0.9399 - 1.0372) |
| | Weighted median | 0.2948 | 0.9712 (0.9196 - 1.0258) |
| | Inverse variance weighted | 0.0440 | 0.9629 (0.9281 - 0.9990) |
| | Simple mode | 0.5846 | 0.9789 (0.9084 - 1.0548) |
| | Weighted mode | 0.2427 | 0.9724 (0.9295 - 1.0172) |
| CD86+ myeloid DC AC | MR Egger | 0.3917 | 0.9778 (0.9303 - 1.0276) |
| | Weighted median | 0.6075 | 0.9858 (0.9333 - 1.0412) |
| | Inverse variance weighted | 0.0389 | 0.9623 (0.9279 - 0.9981) |
| | Simple mode | 0.1616 | 0.9224 (0.8286 - 1.0267) |
| | Weighted mode | 0.1650 | 0.9676 (0.9258 - 1.0112) |
| CD62L- myeloid DC AC | MR Egger | 0.3162 | 0.9550 (0.8776 - 1.0392) |
| | Weighted median | 0.0242 | 0.9301 (0.8733 - 0.9906) |
| | Inverse variance weighted | 0.0111 | 0.9324 (0.8834 - 0.9842) |
| | Simple mode | 0.2275 | 0.9166 (0.8034 - 1.0457) |
| | Weighted mode | 0.0477 | 0.9311 (0.8759 - 0.9897) |
| Resting Treg % CD4 Treg | MR Egger | 0.3070 | 1.0191 (0.9838 - 1.0557) |
| | Weighted median | 0.0184 | 1.0365 (1.0061 - 1.0678) |
| | Inverse variance weighted | 0.0460 | 1.0293 (1.0005 - 1.0589) |
| | Simple mode | 0.8429 | 0.9948 (0.9458 - 1.0464) |
| | Weighted mode | 0.0813 | 1.0252 (0.9983 - 1.0528) |
| Resting Treg %CD4 | MR Egger | 0.0616 | 1.0294 (1.0001 - 1.0597) |
| | Weighted median | 0.0255 | 1.0383 (1.0046 - 1.0732) |
| | Inverse variance weighted | 0.0299 | 1.0260 (1.0025 - 1.0501) |
| | Simple mode | 0.0919 | 0.9870 (0.9256 - 1.0524) |
| | Weighted mode | 0.0180 | 1.0352 (1.0079 - 1.0633) |
| Naive CD8br %T cell | MR Egger | 0.0583 | 0.9901 (0.9804 - 0.9998) |
| | Weighted median | 0.0769 | 0.9898 (0.9786 - 1.0011) |
| | Inverse variance weighted | 0.0451 | 0.9903 (0.9809 - 0.9998) |
| | Simple mode | 0.0399 | 1.0725 (1.0070 - 1.1422) |
| | Weighted mode | 0.0591 | 0.9895 (0.9793 - 0.9999) |
| Monocyte AC | MR Egger | 0.4003 | 0.9847 (0.9508 - 1.0197) |
| | Weighted median | 0.2103 | 0.9761 (0.9397 - 1.0138) |
| | Inverse variance weighted | 0.0271 | 0.9666 (0.9380 - 0.9962) |
| | Simple mode | 0.9896 | 0.9995 (0.9244 - 1.0807) |
| | Weighted mode | 0.1950 | 0.9780 (0.9469 - 1.0100) |
| CD4+ AC | MR Egger | 0.0610 | 0.9336 (0.8748 - 0.9965) |
| | Weighted median | 0.0527 | 0.9331 (0.8700 - 1.0008) |
| | Inverse variance weighted | 0.0389 | 0.9511 (0.9069 - 0.9974) |
| | Simple mode | 0.3772 | 0.9531 (0.8598 - 1.0565) |
| | Weighted mode | 0.0444 | 0.9312 (0.8746 - 0.9916) |
| CD8br %T cell | MR Egger | 0.0848 | 1.0936 (1.0003 - 1.1956) |
| | Weighted median | 0.0753 | 1.0766 (0.9925 - 1.1677) |
| | Inverse variance weighted | 0.0116 | 1.0760 (1.0165 - 1.1390) |
| | Simple mode | 0.1246 | 1.1162 (0.9828 - 1.2676) |
| | Weighted mode | 0.0936 | 1.0791 (0.9965 - 1.1685) |
| CD4+ CD8dim AC | MR Egger | 0.4015 | 1.0915 (0.8992 - 1.3249) |
| | Weighted median | 0.0980 | 1.0883 (0.9845 - 1.2029) |
| | Inverse variance weighted | 0.0276 | 1.0896 (1.0095 - 1.1759) |
| | Simple mode | 0.0792 | 1.1726 (1.0015 - 1.3730) |
| | Weighted mode | 0.1400 | 1.0890 (0.9622 - 1.2073) |

| Traits | Methods | pval | OR (95% CI) |
|---|---|---|---|
| CD4+ CD8dim %lymphocyte | MR Egger | 0.4111 | 1.0707 (0.9160 - 1.2515) |
| | Weighted median | 0.1086 | 1.0766 (0.9838 - 1.1781) |
| | Inverse variance weighted | 0.0355 | 1.0767 (1.0050 - 1.1535) |
| | Simple mode | 0.0661 | 1.1528 (1.0056 - 1.3215) |
| | Weighted mode | 0.1777 | 1.0755 (0.9741 - 1.1876) |
| CD4+ CD8dim %leukocyte | MR Egger | 0.4084 | 1.0705 (0.9165 - 1.2503) |
| | Weighted median | 0.0942 | 1.0800 (0.9869 - 1.1818) |
| | Inverse variance weighted | 0.0362 | 1.0755 (1.0047 - 1.1514) |
| | Simple mode | 0.0566 | 1.1487 (1.0099 - 1.3066) |
| | Weighted mode | 0.1148 | 1.0878 (0.9872 - 1.1985) |
| HLA DR+ CD4+ %lymphocyte | MR Egger | 0.1322 | 1.0988 (0.9808 - 1.2309) |
| | Weighted median | 0.2814 | 1.0480 (0.9623 - 1.1412) |
| | Inverse variance weighted | 0.0438 | 1.0683 (1.0018 - 1.1392) |
| | Simple mode | 0.5319 | 1.0432 (0.9172 - 1.1865) |
| | Weighted mode | 0.3269 | 1.0472 (0.9585 - 1.1441) |
| HLA DR+ NK AC | MR Egger | 0.0744 | 1.1243 (1.0076 - 1.2546) |
| | Weighted median | 0.0165 | 1.0932 (1.0164 - 1.1758) |
| | Inverse variance weighted | 0.0419 | 1.0747 (1.0026 - 1.1520) |
| | Simple mode | 0.1343 | 1.1226 (0.9798 - 1.2863) |
| | Weighted mode | 0.0392 | 1.0967 (1.0190 - 1.1803) |
| CD28- CD127- CD25++ CD8br %T cell | MR Egger | 0.0956 | 1.0132 (0.9506 - 1.0799) |
| | Weighted median | 0.1829 | 1.0431 (0.9803 - 1.1099) |
| | Inverse variance weighted | 0.0359 | 1.0658 (1.0042 - 1.1312) |
| | Simple mode | 0.5137 | 1.0698 (0.8794 - 1.3015) |
| | Weighted mode | 0.2243 | 1.0399 (0.9798 - 1.1038) |
| CD20 on IgD- CD27- | MR Egger | 0.0728 | 0.8398 (0.7057 - 0.9994) |
| | Weighted median | 0.1143 | 0.9249 (0.8395 - 1.0190) |
| | Inverse variance weighted | 0.0425 | 0.9305 (0.8680 - 0.9976) |
| | Simple mode | 0.3258 | 0.9218 (0.7883 - 1.0778) |
| | Weighted mode | 0.2662 | 0.9184 (0.7956 - 1.0602) |
| CD3 on activated & secreting Treg | MR Egger | 0.2813 | 0.9537 (0.8783 - 1.0356) |
| | Weighted median | 0.1440 | 0.9615 (0.9121 - 1.0135) |
| | Inverse variance weighted | 0.0218 | 0.9481 (0.9059 - 0.9923) |
| | Simple mode | 0.0772 | 0.9038 (0.8151 - 1.0022) |
| | Weighted mode | 0.1895 | 0.9634 (0.9138 - 1.0156) |
| CD86 on granulocyte | MR Egger | 0.2589 | 1.0987 (0.9418 - 1.2817) |
| | Weighted median | 0.0590 | 1.0967 (0.9965 - 1.2069) |
| | Inverse variance weighted | 0.0066 | 1.1005 (1.0270 - 1.1793) |
| | Simple mode | 0.3713 | 1.0812 (0.9175 - 1.2741) |
| | Weighted mode | 0.4691 | 1.0603 (0.9098 - 1.2357) |
| CCR7 on naive CD4+ | MR Egger | 0.0226 | 0.9522 (0.9176 - 0.9882) |
| | Weighted median | 0.0688 | 0.9621 (0.9229 - 1.0030) |
| | Inverse variance weighted | 0.0307 | 0.9668 (0.9377 - 0.9969) |
| | Simple mode | 0.1419 | 0.9200 (0.8282 - 1.0219) |
| | Weighted mode | 0.0545 | 0.9605 (0.9249 - 0.9973) |
| CCR7 on naive CD8br | MR Egger | 0.0778 | 1.1329 (1.0003 - 1.2831) |
| | Weighted median | 0.0200 | 1.0953 (1.0144 - 1.1826) |
| | Inverse variance weighted | 0.0135 | 1.0782 (1.0157 - 1.1446) |
| | Simple mode | 0.1281 | 1.1033 (0.9814 - 1.2404) |
| | Weighted mode | 0.0984 | 1.1018 (0.9917 - 1.2241) |
| CD45 on lymphocyte | MR Egger | 0.4461 | 1.0284 (0.9591 - 1.1028) |
| | Weighted median | 0.2075 | 1.0475 (0.9746 - 1.1260) |
| | Inverse variance weighted | 0.0499 | 1.0512 (1.0000 - 1.1050) |
| | Simple mode | 0.0255 | 1.1549 (1.0326 - 1.2917) |
| | Weighted mode | 0.3010 | 1.0365 (0.9710 - 1.1064) |
| CD45 on CD8br | MR Egger | 0.0219 | 1.0469 (1.0128 - 1.0823) |
| | Weighted median | 0.0199 | 1.0484 (1.0075 - 1.0910) |
| | Inverse variance weighted | 0.0201 | 1.0360 (1.0056 - 1.0674) |
| | Simple mode | 0.1836 | 1.0436 (0.9839 - 1.1070) |
| | Weighted mode | 0.0588 | 1.0403 (1.0028 - 1.0791) |
| CD127 on CD28+ CD45RA+ CD8br | MR Egger | 0.1574 | 1.1863 (0.9601 - 1.4658) |
| | Weighted median | 0.0518 | 1.1150 (0.9992 - 1.2444) |
| | Inverse variance weighted | 0.0158 | 1.1061 (1.0192 - 1.2005) |
| | Simple mode | 0.2903 | 1.1105 (0.9262 - 1.3315) |
| | Weighted mode | 0.1950 | 1.1145 (0.9590 - 1.2952) |
| CCR2 on CD14+ CD16+ monocyte | MR Egger | 0.0100 | 0.9612 (0.9354 - 0.9878) |
| | Weighted median | 0.0128 | 0.9618 (0.9328 - 0.9917) |
| | Inverse variance weighted | 0.0036 | 0.9667 (0.9449 - 0.9890) |
| | Simple mode | 0.1141 | 0.9657 (0.9266 - 1.0066) |
| | Weighted mode | 0.0198 | 0.9645 (0.9377 - 0.9920) |
| CD4 on CD45RA+ CD4+ | MR Egger | 0.4525 | 0.9206 (0.7481 - 1.1328) |
| | Weighted median | 0.0533 | 0.9156 (0.8372 - 1.0013) |
| | Inverse variance weighted | 0.0233 | 0.9230 (0.8613 - 0.9891) |
| | Simple mode | 0.0528 | 0.8749 (0.7754 - 0.9871) |
| | Weighted mode | 0.0605 | 0.9092 (0.8316 - 0.9940) |
| CD4 on CD28+ CD4+ | MR Egger | 0.6059 | 0.9584 (0.8201 - 1.1200) |
| | Weighted median | 0.0203 | 0.8994 (0.8224 - 0.9836) |
| | Inverse variance weighted | 0.0233 | 0.9249 (0.8645 - 0.9894) |
| | Simple mode | 0.1130 | 0.8951 (0.7899 - 1.0143) |
| | Weighted mode | 0.1122 | 0.8923 (0.7850 - 1.0144) |
| CD4 on resting Treg | MR Egger | 0.7949 | 0.9708 (0.7837 - 1.2025) |
| | Weighted median | 0.0454 | 0.8840 (0.7834 - 0.9975) |
| | Inverse variance weighted | 0.0269 | 0.8999 (0.8196 - 0.9880) |
| | Simple mode | 0.9957 | 1.0005 (0.8291 - 1.2075) |
| | Weighted mode | 0.1128 | 0.8436 (0.7020 - 1.0139) |

0.5  1  1.5  2

**Fig 2. Forest plots depicting the causal associations between abortion and specific immune cell traits.** \*Abbreviations: IVW, inverse variance weighting; CI, confidence interval.

In Treg cells, Resting Treg % CD4 Treg, Resting Treg %CD4, CD28-CD127-CD25 + + CD8br %T cell and CD127 on CD28+ CD45RA+ CD8br were positively associated with abortion (OR > 1). CD3 on activated & secreting Treg, CD4 on CD28+ CD4+ and CD4 on resting Treg were negatively associated with abortion (OR < 1). In cDC cells, CD62L-HLA DR++ monocyte %monocyte and CD86 on granulocyte were positively associated with abortion (OR > 1). DC AC, CD62L-DC AC, CD62L-DC %DC, CD86+ myeloid DC AC and CD62L-myeloid DC AC were negatively associated with abortion (OR < 1). In the Monocyte group, the following immunophenotypes all have negative causal relationships with abortion: Monocyte AC and CCR2 on CD14+ CD16+ monocyte were negatively associated with abortion (OR < 1). We further confirmed the robustness of this causal relationship using sensitivity analysis (**S1 File** and **S2 Table**). Scatter plot and funnel plot also show the stability of the results (**S2 and S3 Files**).

## Exploration of the causal effect of abortion risk on immunophenotypes

We also conducted a reverse Mendelian randomization analysis to investigate the impact of abortion on immune cell phenotype. The Inverse Variance Weighted method was utilized as the primary approach, with other methods serving as supplementary analyses (**Fig 3** and **S3 Table**). We further confirmed the robustness of this causal relationship using sensitivity

| Traits | Methods | pval | | OR (95% CI) |
|---|---|---|---|---|
| PB/PC %B cell | MR Egger | 0.2492 | | 1.2525 (0.8781 - 1.7866) |
| | Weighted median | 0.2097 | | 1.1700 (0.9155 - 1.4953) |
| | Inverse variance weighted | 0.0277 | | 1.2145 (1.0216 - 1.4438) |
| | Simple mode | 0.6400 | | 1.0942 (0.7599 - 1.5756) |
| | Weighted mode | 0.5079 | | 1.1063 (0.8301 - 1.4745) |
| CD62L- HLA DR++ monocyte %monocyte | MR Egger | 0.0935 | | 0.6868 (0.4664 - 1.0113) |
| | Weighted median | 0.1015 | | 0.8053 (0.6215 - 1.0436) |
| | Inverse variance weighted | 0.0031 | | 0.7507 (0.6208 - 0.9077) |
| | Simple mode | 0.3395 | | 0.8116 (0.5410 - 1.2176) |
| | Weighted mode | 0.2079 | | 0.8116 (0.6003 - 1.0972) |
| CD20 on IgD- CD27- | MR Egger | 0.0722 | | 0.6499 (0.4359 - 0.9689) |
| | Weighted median | 0.2716 | | 0.8633 (0.6642 - 1.1220) |
| | Inverse variance weighted | 0.0432 | | 0.8112 (0.6623 - 0.9936) |
| | Simple mode | 0.5953 | | 0.8896 (0.5878 - 1.3464) |
| | Weighted mode | 0.3855 | | 0.8493 (0.5991 - 1.2038) |
| CD4 on resting Treg | MR Egger | 0.4222 | | 1.1810 (0.8032 - 1.7367) |
| | Weighted median | 0.1849 | | 1.1853 (0.9219 - 1.5240) |
| | Inverse variance weighted | 0.0484 | | 1.2100 (1.0014 - 1.4620) |
| | Simple mode | 0.5445 | | 1.1281 (0.7753 - 1.6414) |
| | Weighted mode | 0.3599 | | 1.1612 (0.8571 - 1.5734) |

0.5    1    1.5    2

**Fig 3. Forest plots depicting the causal associations between specific immune cell traits and abortion.** \*Abbreviations: IVW, inverse variance weighting; CI, confidence interval.

analysis (**S4 Table**). The results showed that abortion could promote PB/ PC% B cell and CD4 on resting the expression of Treg showed a positive causal relationship (p < 0.05, OR > 1). abortion can inhibit CD62L-HLA DR++ monocyte %monocyte and CD20on IgD- The expression of CD27-(P = 0.043,OR = 0.811,95%CI = 0.662~0.993) showed a negative causal relationship (p < 0.05, OR < 1).

## Discussion

Results from our comprehensive Mendelian MR study investigating causal relationships between immune cell phenotypes and abortion risk provide significant insights with profound clinical implications. Our findings unveil a complex interplay among immune cells in the context of abortion, highlighting the critical role of immune regulation in reproductive health.

The identification of specific immune phenotypes that are causally associated with the risk of abortion sheds light on the nuanced role of the immune system in maintaining pregnancy. For example, the positive causal relationship observed in certain B cell phenotypes suggests a potential pathogenic role for these cells in abortion. This finding challenges the traditional view of B cells as primarily protective and emphasizes the need for further research into the mechanisms through which these cells might contribute to adverse pregnancy outcomes [22]. Previous studies have found that B cells have the ability to promote pregnancy tolerance through the release of the immunomodulatory cytokine IL-10 [23]. Our study further revealed that certain subpopulations of B cells are vital for modulating the immune response during pregnancy, which consequently impacts the likelihood of abortion, including Unsw mem %B cell, PB/PC %B cell, IgD+ CD24+ %B cell and Naive-mature B cell %lymphocyte. At present, the mechanism of the role of these B cell subpopulations on abortion are currently unclear, and it is important for future studies to delve deeper into the mechanisms of B cells in immune regulation during pregnancy and how these findings can be utilized for the prevention and treatment of pregnancy complications, particularly abortion.

Conversely, our study also identified negative causal associations in certain T cell maturation stages, including Naive CD8br %T cells, CD4 on CD45RA+ CD4+, and CCR7 on naive CD4+, implying a protective role against abortion. This aligns with existing literature that underscores the importance of T cell-mediated immune tolerance during pregnancy [24]. The intricate balance between different T cell subpopulations and their interactions with other immune cells may be crucial in maintaining fetal-maternal tolerance.

NK cells are an essential element of the innate immune system and have a pivotal role maintaining immune balance at the maternal-fetal interface. Among the 9 TBNK phenotypes associated with abortion found in our study, the remaining 8 TBNK cell phenotypes were pathogenic to abortion, except for In TBNK cell group, where CD4+ AC had a protective effect on abortion. This is consistent with the view that an increase in TBNK cells may induce abortion [25]. However, the mechanism by which TBNK cells cause abortion is not well understood. Previous studies have shown that the cytotoxicity of NK cells is significantly lower in patients who have experienced abortion compared to normal women. It is speculated that this difference may be linked to the cytotoxic effects of NK cells [26].

In addition, our study revealed the association between 7 subsets of CDC cells and 2 immune cell subsets of M cells and abortion. The different immune cell subsets of the same type found in our study had different effects on abortion, suggesting that maintaining a balance in the proportion of immune cell subsets during pregnancy may be important for embryonic growth and maintenance of pregnancy. At the same time, our reverse Mendelian analysis results also revealed that miscarriage can impact certain immune cell phenotypes associated

with B cells, CDC, and Treg cells, indicating an interaction between miscarriage and immune cell phenotypes.

The clinical relevance of our study is multifaceted. Firstly, the immune phenotypes identified could potentially serve as biomarkers for predicting the risk of abortion. This could lead to earlier interventions and personalized management strategies for women at risk. Additionally, understanding the causal mechanisms of immune cells during miscarriage can lead to new opportunities for therapeutic intervention. Intravenous immunoglobulin has been shown to modulate Th1 and Th2 lymphocytes, thereby improving pregnancy outcomes in patients with recurrent miscarriage [27]. Low-dose lymphocyte immunotherapy can rebalance the Th1/Th2/Treg ratio in patients with recurrent miscarriage, thereby benefiting pregnancy maintenance [28]. Therefore, modulation of specific immune pathways or immune cell subsets may be a new strategy to prevent recurrent miscarriage. For instance, modulating specific immune pathways or cell populations may emerge as a novel strategy for preventing recurrent abortions.

However, our study also has certain limitations. The application of genetic data primarily from European populations in our study may limit the generalizability of our findings to other ethnic groups. Furthermore, the complexity of immune system interactions means that our findings represent just one piece of the puzzle. The interaction of genetic, environmental, and lifestyle factors is crucial in determining reproductive outcomes and should be a key area of focus for future research.

In conclusion, our study provides valuable insights into the immunological dimensions of abortion, highlighting specific immune phenotypes that play a causative role. These findings not only deepen our understanding of the immune mechanisms involved in reproductive health but also open the door to innovative approaches in the prevention and management of abortion. Future research, incorporating a more diverse population and multifactorial analysis, is essential to further elucidate these relationships and translate them into clinical practice.

## Conclusion

Our study delves into the causal relationship between different immune cell signatures and abortion through a bidirectional two-sample MR, shedding light on the intricate role of the immune system in pregnancy. And specific immune cell phenotypes may contribute to Abortion. In conclusion, our study opens new avenues for the prevention of Abortion and provides new testing indicators for pregnant women.

## Supporting information

**S1 File.**
(ZIP)

**S2 File.**
(ZIP)

**S3 File.**
(ZIP)

**S1 Table. Estimation of the causal relationship between immune cells and abortion.**
(XLSX)

**S2 Table. Sensitivity analyses for the association between immune cells and abortion.**
(XLSX)

**S3 Table. Full result of reverse MR estimates for the association between immune cells and abortion.**
(XLSX)

**S4 Table. Sensitivity analyses for the association between abortion and immune cells.**
(XLSX)

## Acknowledgments

We extend our heartfelt thanks to all authors for their invaluable contributions to this study.

## Author Contributions

**Conceptualization:** Dan Shen, Jing Cao.

**Data curation:** Wendi Xu, YiZhou Cao, Xinyi Bo, Bing Wen, Fuqiang Zhou, Jing Cao.

**Formal analysis:** YiZhou Cao, Xinyi Bo, Fuqiang Zhou.

**Investigation:** FeiXian Fu, Bing Wen.

**Methodology:** Dan Shen, Jingyi Zheng.

**Resources:** Jing Cao.

**Software:** Jingyi Zheng.

**Supervision:** Dan Shen, Wendi Xu, FeiXian Fu.

**Validation:** Xinyi Bo.

**Visualization:** Jingyi Zheng, Bing Wen.

**Writing – original draft:** Dan Shen.

**Writing – review & editing:** Wendi Xu, Jing Cao.

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
