## [Decision Letter · Decision Letter 0]

24 May 2024

PONE-D-24-01489From genes to reproductive health: immune cell influences on AbortionPLOS ONE

Dear Dr. Fu,

Thank you for submitting your manuscript to PLOS ONE. After careful consideration, we feel that it has merit but does not fully meet PLOS ONE’s publication criteria as it currently stands. Therefore, we invite you to submit a revised version of the manuscript that addresses the points raised during the review process.

**The reviewers raised concerns about how the results and discussions were contextualised. Kindly see their comments and respond accordingingly.**

We look forward to receiving your revised manuscript.

Kind regards,

Chika Kingsley Onwuamah, Ph.D.

Academic Editor

PLOS ONE

Journal Requirements:

2. In the online submission form, you indicated that [The raw data underpinning the conclusions of this article will be made accessible by the authors without undue reservation. For further inquiries, please direct them to the corresponding author.]. 

4. Please upload a copy of Supplementary Table 3 to which you refer in your text on page 10. Please amend the file type to 'Supporting Information'. If the Supplementary file is no longer to be included as part of the submission please remove all reference to it within the text.

Reviewers' comments:

Reviewer's Responses to Questions

**Comments to the Author**

1. Is the manuscript technically sound, and do the data support the conclusions?

Reviewer #1: Yes

Reviewer #2: Partly

2. Has the statistical analysis been performed appropriately and rigorously? 

Reviewer #1: Yes

Reviewer #2: Yes

3. Have the authors made all data underlying the findings in their manuscript fully available?

Reviewer #1: Yes

Reviewer #2: Yes

4. Is the manuscript presented in an intelligible fashion and written in standard English?

Reviewer #1: Yes

Reviewer #2: Yes

5. Review Comments to the Author

Reviewer #1: The manuscript presents a comprehensive study into the causal relationship between immune cell phenotypes and abortion risk using the Mendelian Randomization analysis. Overall, the study presented a rigorous approach to the causal relationship between immune cell phenotypes and abortion risk using Mendelian Randomisation analysis. The authors employed this analysis to address an important area in reproductive health and women's health. Their conclusion demonstrated a complex interplay between immune cells and abortion. They highlighted the critical role of immune regulation in reproductive health and its clinical implications such as immune cell phenotypes being biomarkers for predicting the risk of abortion and also being targets for the development of therapies in reproductive health.

The strengths of the research include;

1. Methodological Rigor: The researchers employed a robust two-sample MR analysis, leveraging large-scale genome-wide association study (GWAS) data and strong sensitivity analyses. This approach enhances the validity and reliability of the findings.

2. Detailed Results: The researchers provided in-depth data on the identified immunophenotypes causally associated with abortion risk. The positive and negative causal relationships are comprehensively described, contributing to a deeper understanding of the mechanisms underlying abortion.

3. Clinical relevance: The study's findings have significant implications for reproductive medicine and maternal health outcomes. Understanding the genetic and immunological factors influencing abortion risk could lead to targeted interventions and improved clinical practices.

Suggestions for Improvement:

1. There is a need for further contextualisation in the introduction section. Although the authors gave a general overview of the research area, the contextualization within the existing literature on reproductive immunology and MR analysis is important for more clarity and relevance.

2.The discussion section could benefit from more thorough interpretation of the findings within the context of existing knowledge and potential clinical implications. Some sections of the manuscript could be further clarified for readability and comprehension by ensuring consistency in terminology and providing clear explanations of statistical methods would improve the manuscript's accessibility to readers. The first sentence in the methodology section referring to schizophrenia needs to be clarified.

Lastly, the conclusion section is ambiguous. The conclusion needs to be succinct. Overall, the manuscript presents valuable contribution to the field of reproductive health, however, minor corrections are recommended to enhance clarity and contextualisation.

Reviewer #2: Review for the manuscript titled “From genes to reproductive health: Immune cell influences on Abortion.”

The role of immunity in risk of miscarriage and recurrent miscarriage has not been in doubt, while the effort put in by the authors in terms of the methods and the statistical analysis is appreciated, the inferences drawn by the authors are not completely consistent with the analysis performed. It is also important for the authors to state how these findings affect the risk of recurrent miscarriage. This should come up under the discussion.

In page 9, under “Materials and methods” and specifically under “Study design”: In the first two statements, the author stated thus “Using two-sample MR Analysis, we evaluated the causal relationship between 731immune cell features (divided into 7 groups) and schizophrenia”. My impression is that this work is on using MR analysis to determine the relationship between immune features and abortion. How did schizophrenia appear in this study?

I understand that data and analysis output from GWAS study are enormous, the authors should however follow this output with summary data in simple tables or figures that will make the message from the analysis clear for interpretation by researchers who intend to benefit from the work.

I am concerned about the approach the authors used in the discussion of the work by referring to another work as if the finding is from this work for instance the authors stated thus “the positive causal relationship observed in certain B cell phenotypes suggests a potential pathogenic role for these cells in abortion [25]”

The preceding statement is linked to the work in reference number 25 and not the authors’ current work. This should be reviewed and worded correctly. The authors should first discuss the findings from their work without bias before comparing to other works irrespective of the original authors of the referenced work. After this is done, the authors can draw inferences from the current work and other previous works.

The authors also repeated this pattern in the following statement and other aspects of the discussion for instance the authors stated thus: “Conversely, our study also identified negative causal associations in certain T cell maturation stages, implying a protective role against abortion [27]”

The conclusion of this work is quite sweeping. The authors should draw specifics from their findings that link the GWAS and the implication it has for the care of women with risk of miscarriage.

The authors should discuss each finding from their analysis exhaustively and compare with other workers and draw credible conclusions from the results. I suggest that the discussion and the conclusions should be written again.

The authors should be careful when referring to other papers to avoid inadvertently transferring statements from other scientific papers to this one.

6. PLOS authors have the option to publish the peer review history of their article (what does this mean?). If published, this will include your full peer review and any attached files.

Reviewer #1: **Yes: **Dr Ijeoma Chinwe Uzoma

Reviewer #2: **Yes: **OHIHOIN AIGBE .G.

---

## [Author Response · Author response to Decision Letter 0]

18 Jul 2024

Dear Editor and Reviewers,

On behalf of all the contributing authors, I would like to express our sincere appreciation of the reviewers’ constructive comments concerning our article entitled “From genes to reproductive health: immune cell influences on Abortion” (ID PONE-D-24-01489).

The feedback provided was highly valuable, contributing to the enhancement of our manuscript. In response to the reviewers' comments, we have amended and supplemented the document, adding further explanations to bolster the credibility of our results. In this revised version, all modifications to our manuscript are highlighted in red text within the file. We address each of the aforementioned issues in detail below.

To Reviewer 1,

1.We fully agree with your view. We have continued to read a substantial amount of literature related to immune cells and miscarriage, providing a more extensive and in-depth background description in the introduction.

2.Thank you for your insightful comments and suggestions. We have thoroughly reviewed the manuscript and corrected any typographical errors. Additionally, we have added more content for discussion, significantly enhancing both the academic rigor and readability of the paper.

3.Thank you for your careful review of the entire manuscript. Based on your suggestions, we have provided a more concise explanation in the conclusion section.

To Reviewer 2,

1.Thank you very much for pointing out this issue. We apologize for the oversight; schizophrenia was a typographical error. We have since removed it.

2.I fully understand your concerns. Following your suggestions, we have meticulously arranged the related content and information in the attachments to facilitate easier reading. We have examined the full text and the associated images. Additionally, we have made modifications to certain parts where the wording was inappropriate.

3.Thank you for your constructive feedback. We have almost entirely rewritten the discussion section and reviewed a substantial amount of literature. This has made our paper more rigorous and academically meaningful. Once again, thank you for your suggestions; they have driven us to improve significantly.

To Editor,

1.Thank you very much for your hard work. We have made revisions according to the format requirements of your journal. If there are any other oversights on our part, please contact us immediately for corrections.

2.Thank you for the reminder. The data on miscarriage is sourced from [https://www.ebi.ac.uk/gwas/publications/34594039]. The data on immune cells is sourced from [https://www.ebi.ac.uk/gwas/publications/32929287]. All data is publicly available for download. Additionally, we have supplemented the data sources in the main text as well.3.We have revised the content according to your formatting requirements.

4.Thank you for the reminder. Supplementary Table 3 was a typographical error, and we have removed it.

---

## [Decision Letter · Decision Letter 1]

6 Aug 2024

From genes to reproductive health: immune cell influences on Abortion

PONE-D-24-01489R1

Dear Dr. Cao,

We’re pleased to inform you that your manuscript has been judged scientifically suitable for publication and will be formally accepted for publication once it meets all outstanding technical requirements.

Kind regards,

Chika Kingsley Onwuamah, Ph.D.

Academic Editor

PLOS ONE

Additional Editor Comments (optional):

Reviewers' comments:

Reviewer's Responses to Questions

**Comments to the Author**

1. If the authors have adequately addressed your comments raised in a previous round of review and you feel that this manuscript is now acceptable for publication, you may indicate that here to bypass the “Comments to the Author” section, enter your conflict of interest statement in the “Confidential to Editor” section, and submit your "Accept" recommendation.

Reviewer #2: All comments have been addressed

2. Is the manuscript technically sound, and do the data support the conclusions?

Reviewer #2: Yes

3. Has the statistical analysis been performed appropriately and rigorously? 

Reviewer #2: Yes

4. Have the authors made all data underlying the findings in their manuscript fully available?

Reviewer #2: Yes

5. Is the manuscript presented in an intelligible fashion and written in standard English?

Reviewer #2: Yes

6. Review Comments to the Author

Reviewer #2: The authors have addressed all the issues raised in my earlier review. The authors have improved upon the discussion of the paper and also addressed the issues I raised regarding the inferences drawn from the results.

7. PLOS authors have the option to publish the peer review history of their article (what does this mean?). If published, this will include your full peer review and any attached files.

Reviewer #2: **Yes: **Ohihoin Aigbe Gregory

---

## [Editor Report · Acceptance letter]

1 Oct 2024

PONE-D-24-01489R1 

PLOS ONE

Dear Dr. Cao, 

I'm pleased to inform you that your manuscript has been deemed suitable for publication in PLOS ONE. Congratulations! Your manuscript is now being handed over to our production team.

Kind regards, 

on behalf of

Dr. Chika Kingsley Onwuamah 

Academic Editor

PLOS ONE